# Cluster size distribution of cells disseminating from a primary tumor

**Mrinmoy Mukherjee**[ID][1]*, **Herbert Levine**[ID][1,2]

**1** Center for Theoretical Biological Physics, Northeastern University, Boston, Massachusetts, United States of America, **2** Depts. of Physics and Bioengineering, Northeastern University, Boston, Massachusetts, United States of America

* mr.mukherjee@northeastern.edu

**Data Availability Statement:** All relevant data are within the manuscript and its Supporting information files.

## Abstract

The first stage of the metastatic cascade often involves motile cells emerging from a primary tumor either as single cells or as clusters. These cells enter the circulation, transit to other parts of the body and finally are responsible for growth of secondary tumors in distant organs. The mode of dissemination is believed to depend on the EMT nature (epithelial, hybrid or mesenchymal) of the cells. Here, we calculate the cluster size distribution of these migrating cells, using a mechanistic computational model, in presence of different degree of EMT-ness of the cells; EMT is treated as given rise to changes in their active motile forces ($\mu$) and cell-medium surface tension ($\Gamma$). We find that, for ($\mu > \mu_{min}$, $\Gamma > 1$), when the cells are hybrid in nature, the mean cluster size, $\overline{N} \sim \Gamma^{2.0}/\mu^{2.8}$, where $\mu_{min}$ increases with increase in $\Gamma$. For $\Gamma \leq 0$, $\overline{N} = 1$, the cells behave as completely mesenchymal. In presence of spectrum of hybrid states with different degree of EMT-ness (motility) in primary tumor, the cells which are relatively more mesenchymal (higher $\mu$) in nature, form larger clusters, whereas the smaller clusters are relatively more epithelial (lower $\mu$). Moreover, the heterogeneity in $\mu$ is comparatively higher for smaller clusters with respect to that for larger clusters. We also observe that more extended cell shapes promote the formation of smaller clusters. Overall, this study establishes a framework which connects the nature and size of migrating clusters disseminating from a primary tumor with the phenotypic composition of the tumor, and can lead to the better understanding of metastasis.

## Author summary

In the process of metastasis, tumor cells disseminate from the primary tumor either as single cells or multicellular clusters. These clusters are potential contributor to the initiation of secondary tumor in distant organs. Our computational model captures the size distribution of migrating clusters depending on the adhesion and motility of the cells (which determine the degree of their EMT nature). Furthermore, we investigate the effect of heterogeneity of cell types in the primary tumor on the resultant heterogeneity of cell types in clusters of different sizes. We believe that the understanding the formation and nature of

**Funding:** This work was supported by National Science Foundation grants PHY-1935762 PHY-2019745 awarded to H.L. The funders had no role in study design, data collection and analysis, decision to publish, or preparation of the manuscript.

**Competing interests:** The authors have declared that no competing interests exist.

these clusters, dangerous actors in the deadly aspect of cancer progression, will be useful for improving prognostic methods and eventually better treatments.

## Introduction

Most of the cancer related deaths (90%) are due to metastasis [1, 2]. In the process of metastasis, cancer cells in the primary tumor recapitulate a conserved developmental program, called the epithelial-mesenchymal transition (EMT), leave the primary tumor, invade the surrounding tissue, travel, enter into the blood vessel (where they are called circulating tumor cells (CTC)), reach distant organs and metastasize. Via EMT, epithelial cells lose their E-cadherin based adhesion and apicobasal polarity, change shape, and gain migratory and invasive nature [3–5]. Although, EMT was originally thought of as a binary transition between epithelial (E) and mesenchymal (M) phenotypes, several recent numerical and experimental studies suggest that there are intermediate stable states (hybrid) possible between E and M states. Specifically, when cells undergo only partial EMT, they partially lose their adhesion properties and also gain migratory capabilities [6, 7], often leading to collective motion.

One of the primary consequence of the collective aspects of hybrid state motility is cluster-based dissemination of cells from the primary tumor, which eventually form detectable CTC clusters. These clusters are small in size (typically 2–8 cells large [8, 9]), and are evidence of a process beyond single cell dissemination of purely mesenchymal phenotype, which form single CTCs. These opposing possibilities are evident from CTC cluster size distributions taken from the patient blood sample [9–14]. Cells clusters have also been observed in pathology images [15, 16] and in several *in vitro* and *in vivo* experiments [17–19]. These CTC clusters have more survival probability and tumor initiation potential as compared to single CTC. By coupling EMT to stemness, it has been observed that the stemness window of cancer stem cells (CSC) lies somewhere in the middle of the EMT spectrum [20–22], or in other words the hybrid states are more likely to become stem like compared to pure epithelial or mesenchymal states and hence hybrid cell clusters have more metastatic potential [23, 24].

Although, a general consensus has been reached relating the hybrid phenotypes and collective migratory clusters of cancer cells [2, 25], the exact mechanism of forming the clusters and their mode of migration are still uncertain [16, 26, 27]. Here, we investigate the effect of adhesion and motility of cells on the cluster size distribution through the formulation of a mechanistic computational model. We associate different degree of EMT-ness to the cells by changing their adhesion and active motile forces; thereafter, we study how the motile cells disseminate from the primary tumor and how the heterogeneity in their EMT-ness [28–30] leads to the heterogeneous clusters [24, 31], which are believed to be more aggressive in terms of plasticity, survival, metastatic potential, and drug resistance [25]. These clusters can be a major contributor to metastasis and hence are a priority for better understanding, improving prognostic methods, and eventually leading to more effective treatment.

## Computational model

We simulate the dynamics of the cells via Cellular Potts Model (CPM) using an open source software CompuCell3D [32]. The simulation box is filled by square pixels (in 2D) of unit area. Each pixel is denoted by a vector of integers $\vec{i}$. The cells are extended objects occupying several pixels. The cell index of the cell occupying pixel $\vec{i}$ is denoted by $c(\vec{i})$ and the type of the cell is denoted by $t(c(\vec{i}))$. Many cells can share the same cell type. The Hamiltonian of the system is

written as:

$$H = H_{contact} + H_{area} + H_{perimeter}$$

where,

$$H_{contact} = \sum_{(\vec{i},\vec{j})neighbors} J_{t(c(\vec{i})),t(c(\vec{j}))}[1 - \delta_{c(\vec{i}),c(\vec{j})}]$$

$$H_{area} = \sum_c \lambda_a(t(c))[a(t(c)) - a_0(t(c))]^2$$

$$H_{perimeter} = \sum_c \lambda_p(t(c))[p(t(c)) - p_0(t(c))]^2$$

$J$ denotes the contact energy or adhesion between two cells, the area and perimeter of the cells are constrained by the targeted area $a_0$ and targeted perimeter $p_0$ respectively, where $\lambda_a$ and $\lambda_p$ set their strength. The empty area of the simulation box which is not filled by the cells, is referred to as medium (m). The medium is treated as a special cell type with unconstrained area and perimeter. These are the standard energy terms used in CPM. In most of the simulations we do not use the $H_{perimeter}$ term.

The dynamics is generated in the system by changing energy ($\Delta H$) with attempts at copying the pixel from the neighboring cells/medium, using a modified Metropolis algorithm, with the probability $P = \{exp(-\Delta H/T : \Delta H > 0; 1 : \Delta H \leq 0)\}$. Here, $T$ is a measure of intrinsic activity of the cells giving rise to cell membrane fluctuations. We sometimes use different temperatures for epithelial cells (denoted as $T_E$) and hybrid cells (denoted as $T_H$); this is done to ensure that the hybrid cells move more effectively than the epithelial ones, as is commonly observed. In such cases, the lower temperature is used at the boundary between cells of different type. The time is tracked by Monte Carlo steps (mcs), where 1 mcs is defined by $n \times n$ pixel copy attempts ($n \times n$ is the total number of pixels in the system). We simulate the system in a 2D square lattice. Nearest-neighbors and second-nearest neighbors have been used to calculate the energy, where the Euclidian distance of the 2nd nearest neighbor from a central pixel is $\sqrt{2}$.

The aforementioned model does not take into account the self-propelled nature of the cells. To take into account the active motility on the cells we incorporate an additional term in the Hamiltonian, $H_{motility}$, defined as [33, 34]:

$$H_{motility} = -\sum_c \mu_c \hat{n}_c.\vec{r}_c$$

where, $\mu$ denotes the motile force acting on the center of mass of the cells and $\hat{n}$ describes the direction of polarity of the cells, which is updated by the rule:

$$\hat{n}_c(time) = \frac{< \vec{v}_c(time)>_{[time-\tau,time]}}{| < \vec{v}_c(time)>_{[time-\tau,time]}|}$$

where, $\tau$ is the persistence time of cell polarity, and $\vec{v}$ is the velocity of a cell's center of mass. We wait $\tau$ steps before updating the polarity of the cells.

At very high value of $T$, $\mu$ and $p_0$, the cells can become fragmented. We work in a part of parameter space where the fragmentation events are very rare. The details of the parameters used in the simulations presented in the different Figures in the main text and the Supporting Information are listed in S1 Table. Finally, sample code is provided in the Github repository

https://github.com/mrinmoy169/Cluster_Size_Distribution.git. All the data presented in this article can be easily generated by using the given parameter set listed in S1 Table.

## Results and discussion

### Cells with diffusive motility

Here, we are interested to see how the H cells are released from the primary tumor. The propensity of cell going towards medium depends on cell-cell (similar and different types) and cell-medium contact energies and the cell motility. In the baseline model without self-propulsion, motility arises from the membrane fluctuations of the cells and hence depends directly on $T$.

Fig 1A represents the starting configuration of a primary tumor in our simulations. The primary tumor is composed of two kind of cells, nonmotile epithelial or E cells (colored in red) and motile hybrid or H cells (colored in green). Our cells have fixed phenotypes throughout, that is we have not as of yet considered in our model an EMT circuit that would enable a cell to switch its status. The number of E (60%), H (40%) cells and total number of cells (here, 517 cells) are fixed throughout the simulations. The H cell positions are initially chosen randomly inside the tumor. The cell-medium interaction is mediated by contact energies or adhesion energies $J_{cell\ medium}$.

To represent the effective propensity towards medium coming from the adhesion or contact energies ($J$), we define three surface tensions ($\gamma$) between E, H cells and medium (m), such as $\gamma_{Em} = J_{Em} - J_{EE}/2$, $\gamma_{Hm} = J_{Hm} - J_{HH}/2$ and $\gamma_{EH} = J_{EH} - (J_{EE} + J_{HH})/2$ (since, $J_{mm} = 0$). We choose a high value of $\gamma_{Em} = 19$ ($J_{EE} = 2$ and $J_{Em} = 20$) and $T_E = 1$ such that E cells remain in contact to each other and behave as a rigid cluster. Now, if $\gamma_{Hm} < 0$, single H cells release from

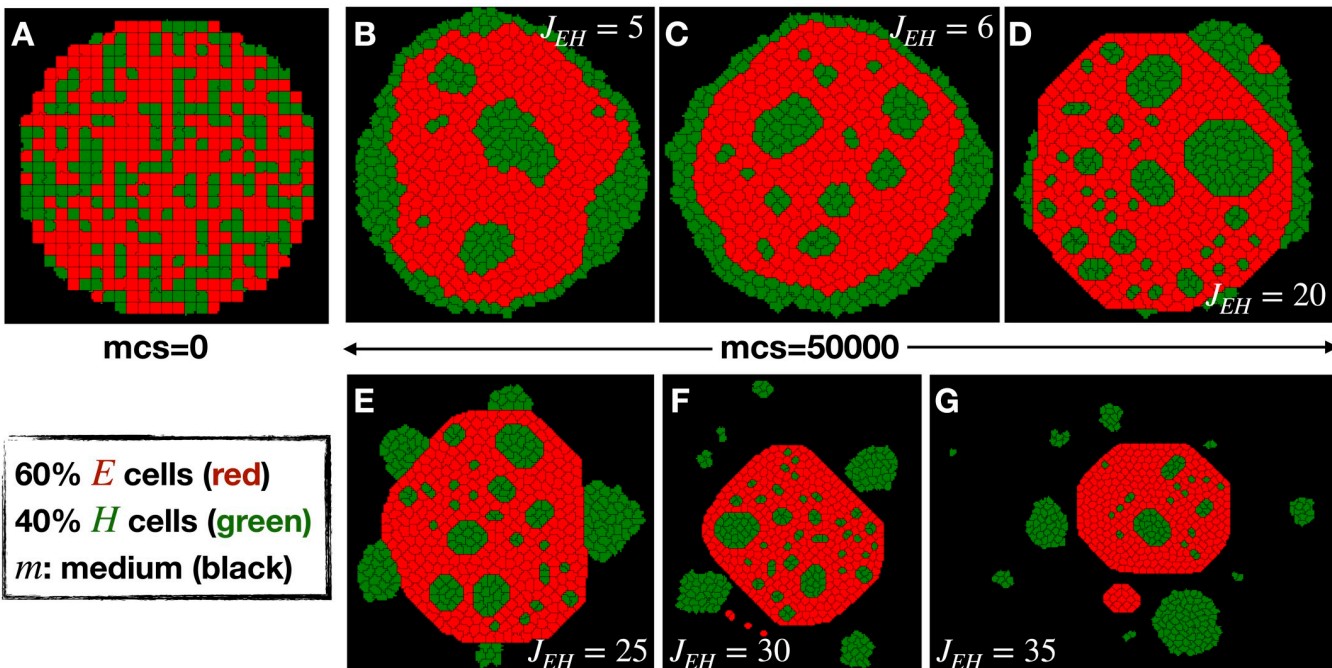

**Fig 1. Dissemination of cells with diffusive motility.** (A) Representative snapshot of initial condition of all the simulations consists of a primary tumor made of 60% nonmotile E cells (red) and randomly chosen 40% motile H cells (green) inside the medium (black) made of no cells. The representative final snapshots of the simulations at mcs = 50000 for different values of $J_{EH}$ = (B) 5, (C) 6, (D) 20, (E) 25, (F) 30 and (G) 35 in case of only fluctuation-based motilities of the cells (in the absence of active motility) at $T_H = 6$.

the tumor, behaving as a mesenchymal cell, because the energy cost of H cells surrounded by medium ($J_{Hm}$) is lower compared to the energy cost of two H cells remaining in contact ($J_{HH}/2$). Therefore, for cluster-based dissemination $\gamma_{Hm}$ should be positive. In the limit $\gamma_{Hm} << \gamma_{Em}$ (here, $\gamma_{Hm} = 3$, $J_{HH} = J_{Hm} = 6$), cells in the tumor will become sorted (Fig 1B) by H cells replacing E cells at the boundary of the tumor (for examples of this type of differential adhesion based cell sorting in the context of CPM see [35, 36]). But to reach dissemination, H cells have to break the bond from the E cells. This can be achieved by increasing the $\gamma_{EH}$ (or equivalently $J_{EH}$, since $J_{EE}$ and $J_{HH}$ are fixed). As shown in Fig 1B–1E, dissemination of H cell clusters can be achieved at a sufficiently high value of $J_{EH}$. Because of this high energy, the H cells inside the tumor can become pinned and the movement of the small H cell clusters inside the E cells drastically slows down. Interestingly, tumor budding occurs (Fig 1E) at intermediate values of $J_{EH} = 25$; budding is a commonly observed feature across different tumors [15, 37].

We can quantify the mean cluster size of H cells in the medium ($\overline{N}$) and fraction of H cells released in the medium ($\overline{f}_H$ = total number of H cells released in the medium/total number of H cells in the system) as a function of $J_{EH}$ and $T_H$. $\overline{N}$ and $\overline{f}_H$ increases with the increase in $T_H$ at a fixed $J_{EH}$, but do not vary with the $J_{EH}$ at a fixed $T_H$ (see S1 Fig). We can conclude that, due to the presence of fluctuation-based motility, having ($T_H$), higher value of $J_{EH}$ helps to break the bonds between E and H cells. The value $T_H$ controls the formation of clusters and hence their sizes and released fraction.

If there is a high probability of cells becoming pinned in the interior, why does EMT seem to invariably lead to dissemination? The issue of pinning of the cells inside the tumor can be overcome by the fact that the EMT inducing signals (which convert E cells to H cells) actually tend to act on the periphery or the invasive front of the tumors [38]. So, the probability of creating H cells inside the tumor is modest and hence there is no real problem with some residual cell pinning in the interior of the tumor. Even if EMT mostly happens in the interior of the tumor, motile hybrid cells (for $\gamma_{Hm} > 0$) can reach the the periphery of the tumor as long as $J_{EH} < J_{HH}$ (otherwise all small clusters will be pinned inside the tumor). So, we can imagine a architecture of the primary tumor composed of epithelial (E) cells ($\gamma_{Em} >> 0$) in the interior, hybrid (H) cells in the periphery ($0 < \gamma_{Hm} < \gamma_{Em}$) and mesenchymal (M) cells ($\gamma_{Mm} < 0$) in the medium [39].

## Cells with active motility

So far, we have seen that H cells can leave the primary tumor only at large $J_{EH}$. But, this result neglects one extremely important feature of H cells, namely that they have become actively motile undergoing (partial) EMT. Thus, a convincing way to overcome having to choose a high value of $J_{EH}$ for cluster based dissemination is to introduce active motility for the H cells. For this purpose, we choose $J_{EH} = max(J_{EE}, J_{HH}) = J_{HH}$ and $T_H = 1$ (similar to $T_E$) and add an active motility term in the Hamiltonian for the H cells, in a similar fashion to that described in the literature [34, 40, 41]). The velocity of the individual cells and their cooperativity with neighboring H cells can then be controlled by the motile force coefficient ($\mu$ as defined above), the persistence time of cell polarity ($\tau$) (see [42, 43]) and surface tension between the H cells and the medium ($\gamma_{Hm} \equiv \Gamma$). When a cluster of H cells are aligned in a direction towards the medium from the tumor, the effective force will enable them to pull out from the primary tumor and thereby an H cell cluster is disseminated. If a very large cluster is released, some cells in this cluster may rearrange and polarize collectively in a different direction as compared to the other cells. As time progresses, this leads to the cluster breaking into smaller clusters; this occurs the same way the initial large cluster breaks from the primary tumor. Obviously, these breaking events depend on the detailed balance of $\mu$, $\Gamma$ and $\tau$ for the H cells. A typical

demonstration of these two-step breaking events is shown in S2 Fig, by plotting the velocity map of the cells near the primary tumor.

Let us consider a sufficiently large value of $\tau = 50$ (we will discuss below the effect of varying $\tau$) for clusters to maintain coordinated motility. We can then investigate the effect of $\mu$ and $\Gamma$ on the cluster size distributions of H cells. In almost all cases, the clusters are composed of only H cells migrating through the medium. Sometimes, a small number E cells exits the primary tumor along with the H cells from the primary tumor, but these cells quickly separate out from the H cells and remain near the primary tumor due to their immobile nature; we will ignore those very few clusters of E cells. We see (S3 Fig) the initial increase of mean cluster size ($\overline{N}$) of H cells due to the primary dissemination from the E cells of the tumor and eventually the decrement with time (mcs) due to the breakup, discussed earlier and finally the saturation across different $\mu$ and $\Gamma$. At the time when $\overline{N}$ reaches to a maximum at a particular ($\mu$, $\Gamma$), the maximum possible number of H cell release in the medium. To account for these multiple breaking and merging events, we wait for long time (here 50000 mcs), such that the $\overline{N}$ reaches a saturation. We calculate the statistical quantities at mcs = 50000 by averaging over many (20–30) independent simulations. One limitation on this time arises due to the limited size of the simulation box and periodic boundary condition; in particular, at very high values of $\mu$ and/or low value of $\Gamma$ rapidly moving cells/clusters cross the boundary of simulation box and enter into the box from opposite side, hit other cells/clusters.

Fig 2A and 2B depicts the mean cluster size of H cells ($\overline{N}$) in the medium and the fraction of H cells ($\overline{f}_H$) released into the medium respectively, as a function of the motile force strength ($\mu$) for different values of H cell-medium surface tension ($\Gamma$). For $\Gamma < 0$, we see single H cell ($\overline{N} = 1$) dissemination irrespective of the motile forces ($\mu$). As we discuss earlier in case of only fluctuation-based dissemination, for $\Gamma < 0$ the cells actually prefer the medium and hence leave as single cells. The number of cells released actually depends on the strength of $\mu$. As shown in Fig 2B, $\overline{f}_H$ increases gradually with increase in $\mu$ and starts to saturate at $\mu = 30$. A similar thing happens for $\Gamma = 0$ as well. Only at $\mu = 0$ it is very difficult to release H cells. Release also depends on the fluctuations (here, $T_H = 1$) and bulk modulus (the spring constant for the area constraint, here $\lambda_a = 1$) of the cells. As shown in S4(A) and S4(B) Fig, $\overline{N}$ gradually decreases with increase in cell membrane fluctuation $T_H$ at a particular ($\mu$, $\Gamma$), whereas $\overline{f}_H$ remains constant. So, $T_H$ actually affects the break-up of the clusters in the medium. The modulation of $\overline{N}$ and $\overline{f}_H$ with the change in $\lambda_a$ is complicated (see S4(C) and S4(D) Fig). For very small value of $\lambda_a$, the cells become more flexible and it is difficult to maintain large clusters, hence the decrease in $\overline{N}$. In the other limit, at very high value of $\lambda_a$, the cells become very rigid and it is difficult to release the cells to the medium.

For $\Gamma > 0$, we note that there is a minimum value $\mu$ ($\mu_{min}(\Gamma)$, since the other parameters are fixed), to release cells and also the standard deviations in $\overline{N}$ increases as we approach these $\mu_{min}$. The $\mu_{min}$ increases with increasing $\Gamma$. The mean cluster size gradually decreases with increase in $\mu$ and/or decrease in $\Gamma$. More cells release as we increase $\mu$ and/or decrease $\Gamma$ of the cells. Even at $\mu = 50$, a few H cells (10–15%) remain in the primary tumor. This residual number, which may affect the extent to which the primary tumor can exhibit mesenchymal gene expression patterns, can be controlled by changing the rigidity (E cell-medium surface tension $\gamma_{Em}$) of the E cells or by altering the E-H bond strength (E cell-H cell surface tension $\gamma_{EH}$). We note that $\overline{f}_H$ gradually increases with the decrease in $\gamma_{Em}$, but $\overline{N}$ remains unchanged (see S5(A) and S5(B) Fig). At very small value of $\mu$ and/or large value of $\Gamma$, we can see a slight increment in $\overline{N}$ with the decrement in $\gamma_{Em}$. In this limit (low $\mu$, high $\Gamma$, high $\gamma_{Em}$) there are not enough cells likely to release in the medium. So, lower value of $\gamma_{Em}$ actually can help the H cells to

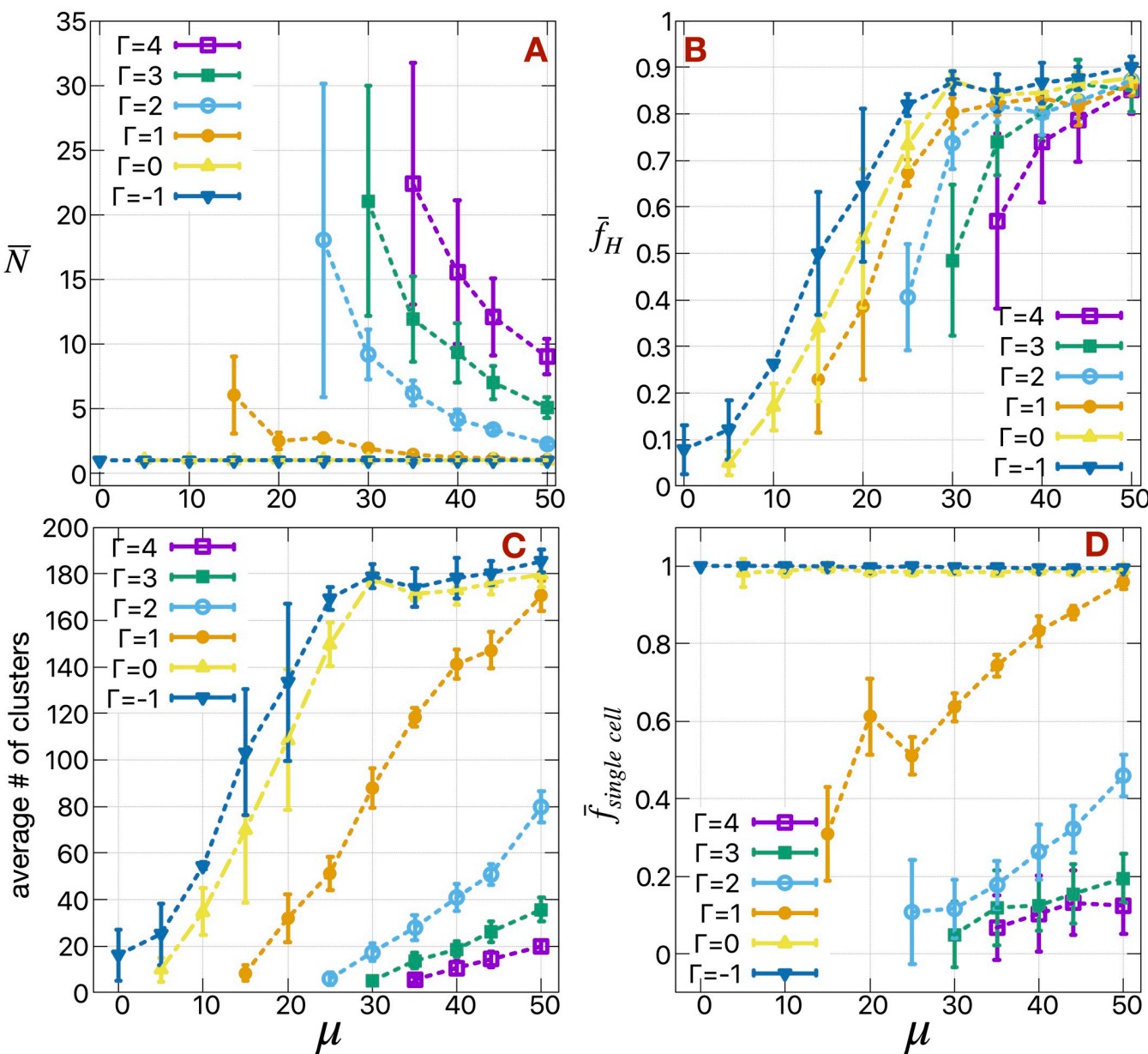

**Fig 2. The mean cluster size, the fraction of motile cells in the medium, the average number of clusters and the fraction of single cell clusters as a function of motile forces and cell-medium surface tension.** (A) The mean cluster size ($\overline{N}$) of H cells in the medium, (B) the fraction of H cells ($\bar{f}_H$) released in the medium, (C) mean number of H cell clusters in the medium and (D) fraction ($\bar{f}_{single\ cell}$) of single cells ($N = 1$) in the total number of H cell clusters as a function of motile forces ($\mu$) of the H cells for different values of surface tension between H cells and medium (m) $\gamma_{Hm} \equiv \Gamma$. We use the standard deviations of the simulated data as error bars.

break the bond between E-H and release as a large clusters. Otherwise, $\overline{N}$ solely depends on ($\mu$, $\Gamma$). The effect of $\gamma_{Em}$ also manifests in the shape of the primary tumor during simulations (see S5(E)–S5(H) Fig), as it becomes more roundish at lower value of $\gamma_{Em}$. We do not see any significant effects on $\overline{N}$ or $\bar{f}_H$ with the change in intrinsic fluctuations ($T_E$) of the E cells (see S5 (C) and S5(D) Fig). The remaining cells, which do not release, form tumor buds as discussed previously.

The decrement of $\overline{N}$ with the increment in $\mu$ and/or decrement in $\Gamma$ is also manifested in the mean number of H cell clusters in the medium (Fig 2C) and fraction ($\overline{f}_{single\ cell}$) of single cells ($N = 1$) in the total number of H cell clusters (Fig 2D). The mean number of clusters and the number of single H cells in the medium increase with increase in $\mu$ and/or decrease in $\Gamma$. Obviously, $\overline{f}_{single\ cell} = 1$ irrespective of $\mu$ in case of $\Gamma = -1$ and 0. As a consequence of getting a large number of disseminated H cell clusters, the mean cluster size decreases and the number of single H cells increases. As we believe, these clusters eventually form circulating tumor cells (CTC) in the blood stream. It has been observed in case of head and neck cancer, patients with higher total CTC count have a lower percentage of multicellular CTC clusters or a higher percentage of single cell CTCs [44].

## Scaling behavior

It is reasonable to assume that different H cells (perhaps in different tumor types) could exhibit different degrees of active motility; hence it is interesting to study how our results change as $\mu$ is varied. We observe similar behavior for the trends of $\overline{N}$ with varying $\mu$ in Fig 2A, at least in the range $\Gamma = [2, 4]$. This suggests that a rescaling of the axis could lead to data collapse. Indeed, the $\overline{N}$ data does collapse into a master curve in the range $\Gamma = [2, 4]$ and for higher $\mu$ in case of $\Gamma = 1$ as well, when we normalize it with respect to $\overline{N}_{\mu=50}$ (Fig 3A). For $\Gamma = [-1, 0]$ all the $\overline{N} = 1$, and $\Gamma = 1$ actually separates out the two range of $\Gamma$ $[-1, 0]$ and $[2, 4]$ in terms of $\overline{N}$. Next, we fit the collapsed data to a power law $(50/\mu)^n$, and find $n = 2.8 \pm 0.3$. The normalization factor $\overline{N}_{\mu=50}$ previously used, depends on the $\Gamma$. As shown in Fig 3B, we fit $\frac{\overline{N}_{\mu=50}}{\overline{N}_{\mu=50}(\Gamma=2)}$ to a power law $(\Gamma/2)^m$, and find $m = 2.0 \pm 0.2$. Combining these two relations, we get $\overline{N} \approx 3.2 \times 10^4 \left(\frac{\Gamma^{2.0}}{\mu^{2.8}}\right)$. Interestingly, the rate of change in $\overline{N}$ with respect to $\mu$ (power $\approx -2.8$) is more rapid compared to that with $\Gamma$ (power $\approx 2.0$).

We verify this relation by plotting $\overline{N}$ as a function of $x = \Gamma^{2.0}/\mu^{2.8}$ (Fig 3C) and recovers the linear relationship by fitting the data in two ways, by a straight line $N_0 x$ (we get $N_0 = 35329 \pm 1434$, which confirms the slope of the linear relationship) and by a power law $32470x^p$ (we get $p = 0.99 \pm 0.01$, which confirms the linearity of the relationship). The data fit better in the lower range of $x$ (equivalently the higher range of $\mu = [40, 50]$ and/or lower range of $\Gamma = [1, 3]$). The noise in the data also increases for $\mu < 40$ and/or $\Gamma > 3$ and $\overline{N}$ becomes rather high ($> 10$). For comparison, the CTC clusters seen experimentally are typically 2–8 cells large [8, 9].

Similarly, we can fix $\Gamma = 3$ and plot $\overline{N}$ as a function of $x = \Gamma^{2.0}/\mu^{2.8}$ (Fig 3D) as in Fig 3C for different values of persistence time of cell polarity and for $\mu = [40, 50]$. As shown in S6 Fig, both the $\overline{N}$ and $\overline{f}_H$ gradually increases with increase in $\tau$ and starts to saturate at $\tau = 50$ for all the values of $\mu$. Higher value of $\tau$ actually helps the cell to move more cooperatively, and hence maintain larger clusters. We choose $\tau = 50$ for the rest of the article.

Until now, we have discussed the mean cluster size of H cells. S7(A)–S7(F) Fig depict the cluster size distributions for different values of $\Gamma$ and $\mu$. The probability of getting large clusters ($N > 10$) is very small ($< 5\%$) irrespective $\mu$ or $\Gamma$, and the distributions fall rapidly as $N$ increases up to $N = 10$. The fall-off rate of these distributions depends on ($\mu$, $\Gamma$). As discussed before for $\Gamma \leq 0$, we find only single H cells in the medium (S7(E) and S7(F) Fig). The probability of getting single cells increases with the increase in $\mu$ and/or decrease in $\Gamma$ (also shown in Fig 2D).

For comparison with these model results, we show cluster size distribution data collected from several articles [9–12, 14], as combined in [45]. Note the similarity in the distributions of CTC clusters across different cancer types, as shown in Fig 4A–4F. As we increase $\Gamma$ or decrease $\mu$, the probability of getting smaller clusters ($N = 2$) decreases. Moreover, at very high

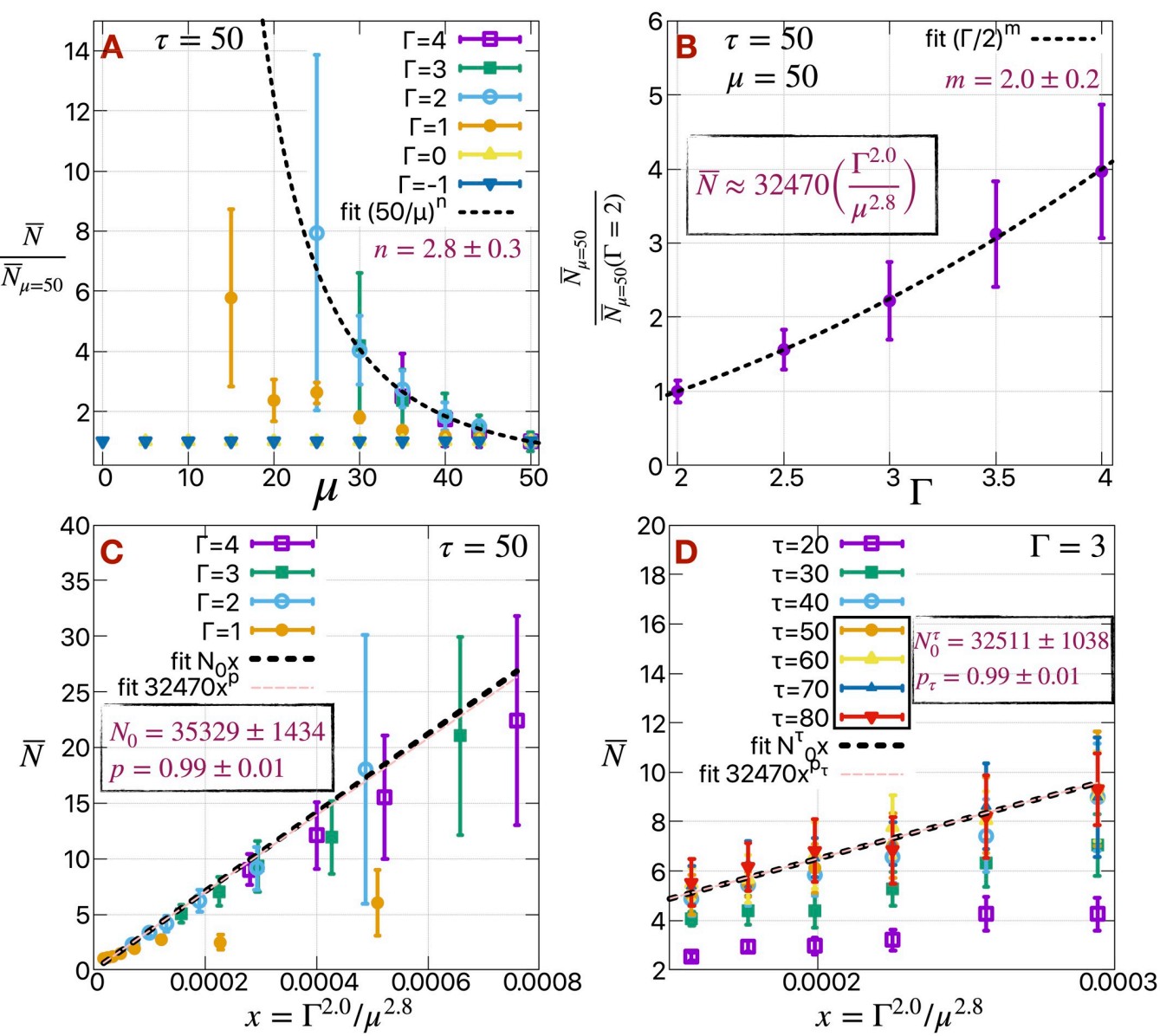

**Fig 3. Scaling behavior of mean cluster size.** (A) $\frac{\overline{N}}{\overline{N}_{\mu=50}}$ as a function of $\mu$ representing the collapse of all the data shown in Fig 2A into a master curve for $\Gamma > 1$.
(B) $\frac{\overline{N}_{\mu=50}}{\overline{N}_{\mu=50}(\Gamma=2)}$ as a function of $\Gamma$. (C) $\overline{N}$ as a function of $x = \Gamma^{2.0}/\mu^{2.8}$. (D) $\overline{N}$ as a function of $x = \Gamma^{2.0}/\mu^{2.8}$ for different values of persistence time of cell polarity ($\tau$).
The details underlying the fitting of these data are described in the main text.

value of $\Gamma$ and small value of $\mu$ the peak of the distribution shifts from $N = 2$ (Fig 4F); the probability of getting intermediate size clusters ($N > 2$) increases.

We also checked the effect of larger system size by choosing a primary tumor with a larger number of cells ($n_E = 1265$, $n_H = 844$) and compare the time series of mean cluster size ($\overline{N}$) with that of our previous system ($n_E = 310$, $n_H = 207$). Due to availability of large number of $H$ cells in the larger tumor, the clusters released initially are bigger compared to the clusters released from the smaller tumor; eventually, however, the $\overline{N}$ reach a similar value corresponding to their ($\mu$, $\Gamma$), as they break apart in the medium (S8(A) and S8(B) Fig). Also, note the

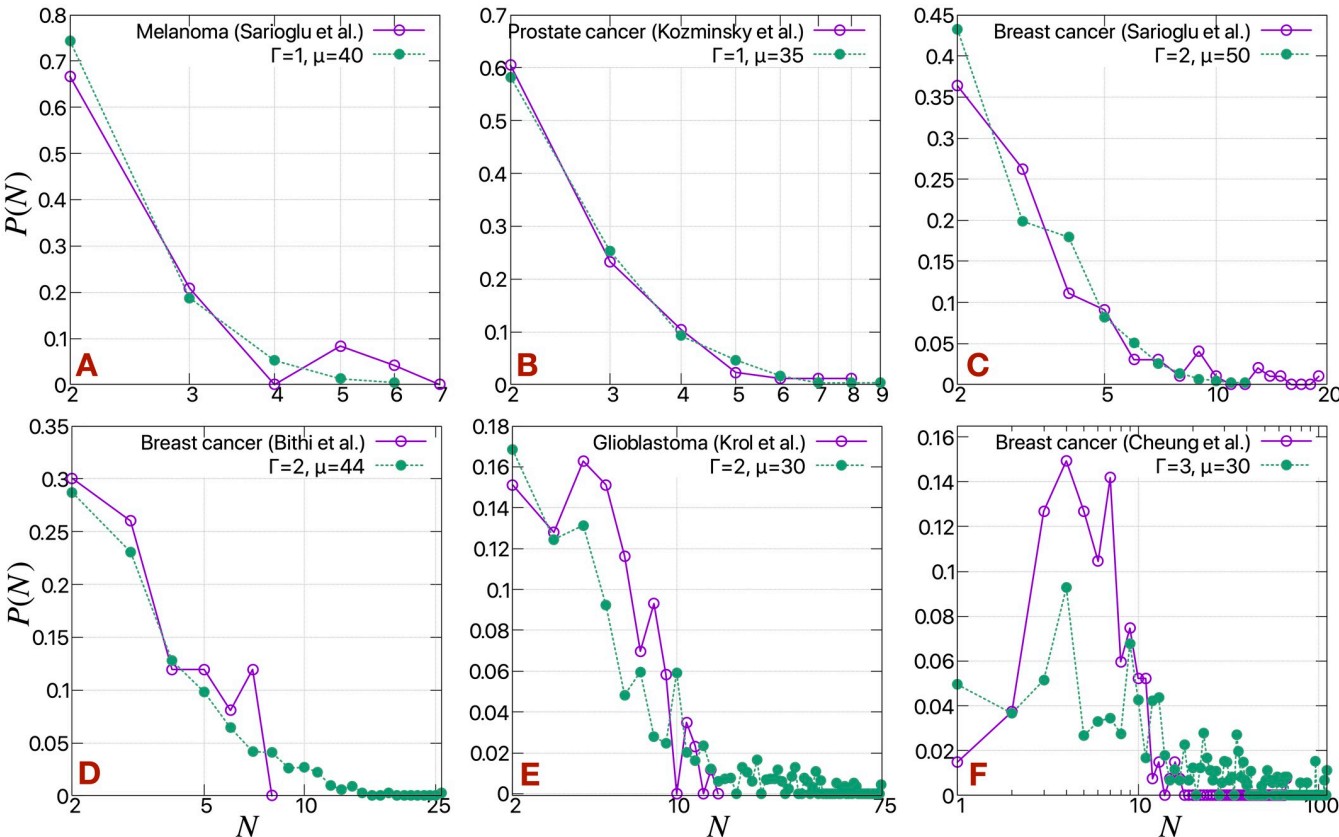

**Fig 4. Cluster size distributions compared with experimental results.** Cluster size distributions ($P(N)$) of H cells for different values of $\Gamma$ and $\mu$ compared with the circulating tumor cell (CTC) cluster size distributions taking from different experiments. The distributions are normalized such that $\Sigma\, P(N) = 1$. The $N$ axes are in log scale.

similarity in the cluster size distributions at mcs = 50000 (S8(C) and S8(D) Fig). Note that due to the periodic boundary conditions, some cells may cross the boundary of the simulation box and enter into the box from opposite side, hit and merge with other cells, it is therefore important to increase the size of the simulation box for larger tumor to maintain the number density of $H$ cells fixed. Otherwise, $\overline{N}$ varies (slightly) according to the number of cells in the tumor at mcs = 50000.

In summary, the cells with lower value of $\mu$ and/or higher value of $\Gamma$ are more epithelial-like, and promote the existence of larger clusters; whereas in the other limit at higher value of $\mu$ and/or lower value of $\Gamma$, the cluster size decreases. For $\Gamma > 0$, $\mu$ should be $> \mu_{min}$ to release cells and for $\Gamma \leq 0$, the cells will leave individually.

## Heterogeneity of cell types in the clusters

Heterogeneity of cell phenotypes is a common feature of different tumor cell populations, in the primary tumor [25, 28–30], in secondary tumor in distant organs [38, 46] and in CTC clusters [24, 31]. In other words, we should expect significant variability in the degree of EMT that different cells have undergone. It will thus interesting to see how variability, in terms of motile forces and cell-medium surface tension, of cells in the primary tumor can affect the cluster size distributions and also determine the composition of different cell types in these clusters.

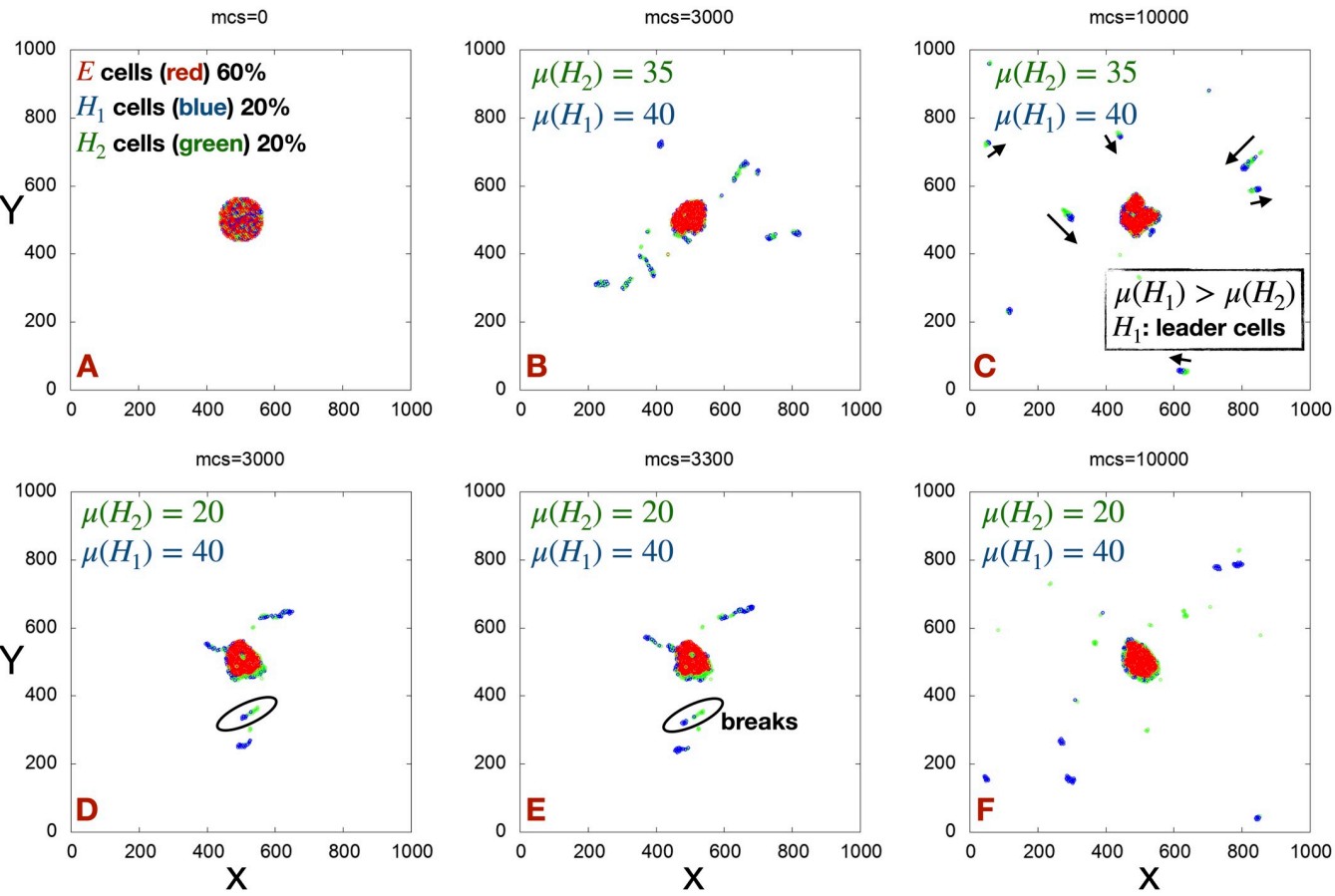

**Fig 5. Snapshots of heterogeneous clusters.** (A) A representative snapshot of an initial configuration consisting of two kinds of H cells ($H_1$ and $H_2$) differing in their motile forces ($\mu$). Representative snapshots of the simulations at different mcs for (B-C) $\mu(H_1) = 40$, $\mu(H_2) = 35$ and (D-F) $\mu(H_1) = 40$, $\mu(H_2) = 20$ with a fixed $\Gamma = 3$ for all the H cells. The arrow indicates the direction of movement of the clusters. Observe the cluster breaking event occurring inside the black circles. Here, we have plotted the center of mass of the cells.

To start, we consider having two kinds of H cells, $H_1$ and $H_2$, with the number ratio of cells initially in the primary tumor chosen to be $E : H_1 : H_2 = 6 : 2 : 2$; these H cells differ in their motile forces ($\mu$) at a fixed cell-medium surface tension ($\Gamma = 3$). We choose $\mu(H_1) = 40$ and vary $\mu(H_2)$. As shown in Fig 5A–5C and S1 Video for $\mu(H_2) = 35$, cells with higher motility (blue $H_1$ cells) act as leader cells, determining the direction of movement of the heterogeneous clusters composed of both $H_1$ (blue) and $H_2$ (green) cells. This phenomenon of leader-follower cells has been observed in many examples of collective cell migration [47].

In a case of larger motility difference, specifically for $\mu(H_2) = 20$ (see Fig 5D–5F and S2 Video) $H_1$ cells acting as a leaders help $H_2$ cells to exit the primary tumor. Note that $H_2$ cells cannot disseminate alone without the help of higher motile $H_1$ cells, since for $\Gamma = 3$ the minimum motile forces required to disseminate cells $\mu_{min} > 20$). This finding is somewhat analogous to experimental cases where non-cancerous cells like fibroblasts can help to release the cells with lower motility (more epithelial like) [40, 48]. Although the $H_1$ cells help $H_2$ cells to disseminate, they cannot maintain contact with the $H_2$ cells due to their high difference in motile forces. So, at long time the probability of seeing heterogeneous clusters (composed of both the cells) decreases and most of the clusters become homogeneous.

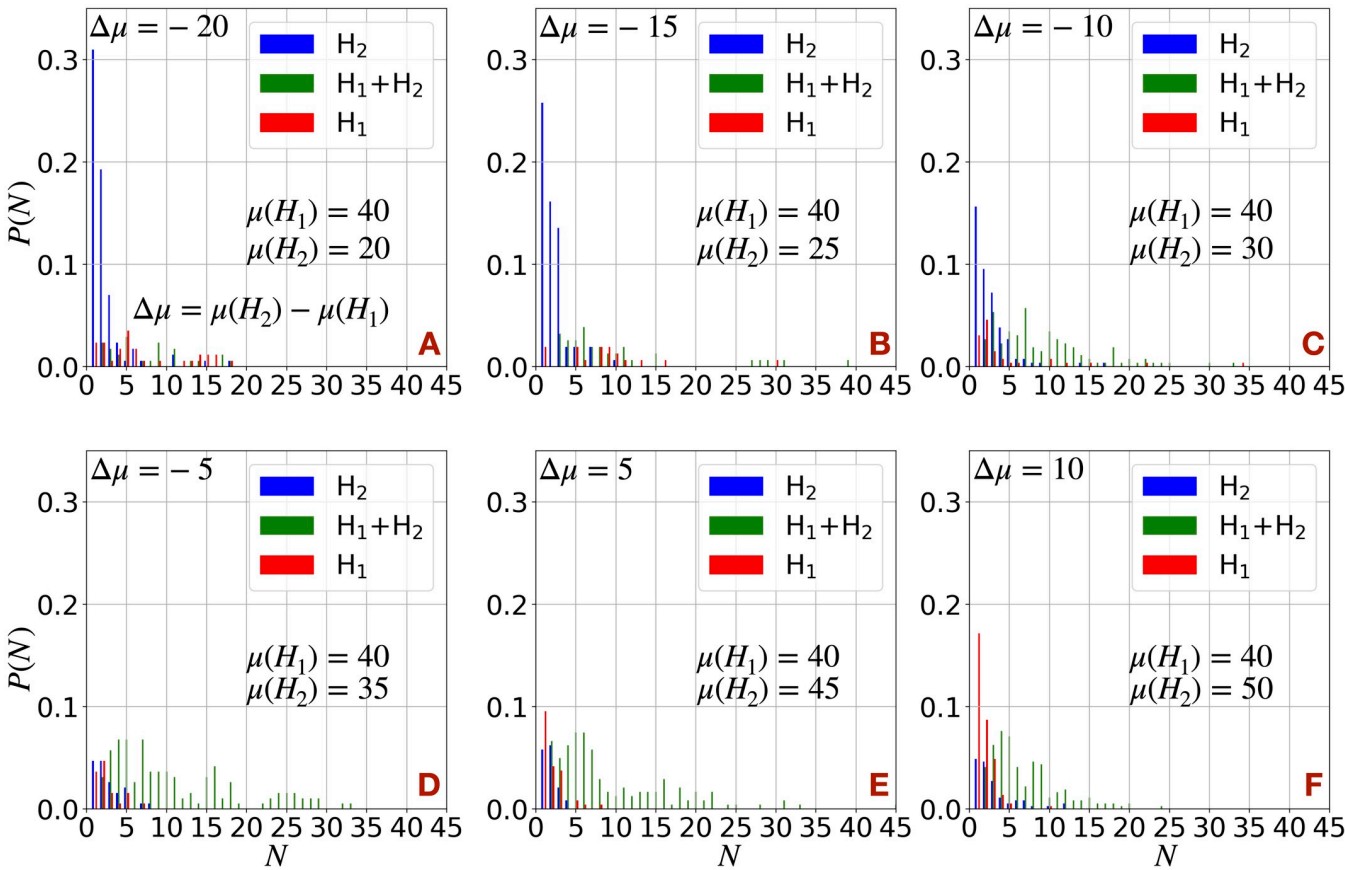

**Fig 6. Cluster size distributions in case of heterogeneous clusters.** The cluster size distributions ($P(N)$) of H cells in case of two kinds of H cells different in their motile forces at a fixed $\Gamma = 3$ for different values of $\Delta\mu = \mu(H_2) - \mu(H_1)$ by changing $\mu(H_2)$ keeping fixed $\mu(H_1) = 40$. At each $N$, $P(N)$ is split into three bars. Red bars represent the clusters consist of only $H_1$ cells, blue bars represent the clusters consist of only $H_2$ cells and green bars represent the heterogeneous clusters consist of $H_1$ and $H_2$ cells. The distributions are normalized such that $\Sigma\, P(N) = 1$.

In Fig 6 we calculate the cluster size distributions ($P(N)$) of the cells in the medium at mcs = 50000 across different values of $\Delta\mu$, defined as $\Delta\mu = \mu(H_2) - \mu(H_1)$, by changing $\mu(H_2)$ keeping $\mu(H_1)$ (=40) fixed. At each cluster size ($N$), $P(N)$ is splitted into three bars, red bars represent the clusters consist of only $H_1$ cells, blue bars represent the clusters consist of only $H_2$ cells and green bars represent the heterogeneous clusters consist of $H_1$ and $H_2$ cells. The probability of getting heterogeneous clusters (green bars) decreases with increase in $|\Delta\mu|$. The effects are somewhat symmetric for both positive ($\mu(H_1) < \mu(H_2)$) and negative ($\mu(H_1) < \mu(H_2)$) $\Delta\mu$. Asymmetry in mean cluster size ($\overline{N}$) arises for lower value of $\mu(H_1)$ (see S9(A) and S9(B) Fig). For example, $|\Delta\overline{N}|(\Delta\mu = -5) >> |\Delta\overline{N}|(\Delta\mu = 5)$ at $\mu(H_1) = 30$, where $\Delta\overline{N}$ is defined as, $\Delta\overline{N} = \overline{N}(H_2) - \overline{N}(H_1)$. But, this difference in $|\Delta\overline{N}|$ decreases with the increase in $|\Delta\mu|$. This is the manifestation of combination of three effects– (1) the mean cluster size decreases with the increase in $\mu$ (as observed in one type of H cell cases), (2) for very small value of $\mu$ when the cells can not come out easily from the primary tumor, the cells with higher $\mu$ take out a smaller portion of the cells with lower $\mu$ and (3) when $|\Delta\mu|$ is high the heterogeneous clusters composed of two kind of cells break.

Interestingly, we observe that in case of high $|\Delta\mu|$ ($> 5$), single cells or very small clusters in the medium are largely composed of the cells with lower motility; see the blue bars for $\Delta\mu < 0$

and red bars for $\Delta\mu > 0$ in Fig 6. This result is opposite to the intuition that cells with higher motility produce smaller clusters; for example, we saw previously in case of one type of H cell, that the probability of getting small clusters increases with the increase in $\mu$. Actually, in presence of both $H_1$ and $H_2$ cells, the cells with higher motility guide the primary dissemination of the cells with lower motility. Afterwards, at high value of $|\Delta\mu|$ heterogeneous clusters, which have a greater proportion of highly motile cells on average, breaks. Consequently, cells with higher motility maintain the larger clusters by leaving behind the smaller clusters composed of cells with lower motility. It would clearly be interesting to perform dissemination experiments in the presence of many cells exhibiting different degrees of EMT.

Next, we check the effect of heterogeneity in cell-medium surface tension $\Gamma$. For this purpose, we again consider two kinds of H cells now differing in $\Gamma$ at a fixed $\mu$. As shown in S9(C) and S9(D) Fig, and opposite to the previous case, $|\Delta\overline{N}|$ decreases with an increase in $|\Delta\Gamma|$ ($\equiv \Delta\Gamma = \Gamma(H_2) - \Gamma(H_1)$) in case of $\Delta\Gamma < 0$ but remains the same (or slightly increases) with increased $|\Delta\Gamma|$ in case of $\Delta\Gamma > 0$. The probability of obtaining heterogenous clusters (S10 Fig) is also similar (with no clear trend) across different values of $|\Delta\Gamma|$ except for $\Delta\Gamma = -2$, in this case $\Gamma(H_2) = 1$, which promotes single cell dissemination for $\mu = 30, 40$ (we saw this previously when we had one type of H cell). One of the reason for different trends in $|\Delta\overline{N}|$ with respect to $|\Delta\mu|$ and $|\Delta\Gamma|$ can be the range of these $\Gamma$ values, $|\Delta\Gamma|$ is not high enough (for $\Delta\Gamma > 0$) to see significant effects on $|\Delta\overline{N}|$, comparable to what we saw for $|\Delta\mu|$. The rate of change in $\overline{N}$ is slower with respect to the change in $\Gamma$ ($\sim \Gamma^{-2.0}$) compared to $\mu$ ($\sim \mu^{-2.8}$). Therefore, to see the effect of very high value of $\Gamma$, we would have to shift the range of $\mu$ towards higher values. So, for the remainder of this work, we vary $\mu$ instead of $\Gamma$ to mimic the change of the EMT status of the cells.

## A spectrum of EMT states

In general, there can be a wide spectrum of EMT states present in the primary tumor. Using our computational model, we can address this case by including cells with a spectrum of $\mu$ values at a fixed $\Gamma$, assuming that the different degrees of EMT of the cells manifests itself mostly in their active motility. To investigate this scenario, we choose random $\mu$ values for all the H cells, drawn from a normal distribution with mean 40 and standard deviation (std) $\sigma$ all at $\Gamma = 3$. Fig 7A depicts the mean cluster size of the cells released in the medium ($\overline{N}$) as a function of $\sigma$. $\overline{N}$ remains roughly constant up to $\sigma = 2$, above which it gradually decreases with increased $\sigma$. This change is perhaps more evident from the cluster size distributions shown in Fig 7B, as $\sigma = 2$ clearly separates out two distinct trends in $P(N)$ with respect to $N$– (1) Specially, for $\sigma \leq 2$, $P(N)$ starts to fall rapidly as a function of $N$ after $N > 3$ and (2) for $\sigma > 2$, $P(N)$ falls rapidly as a function of $N$ starting from $N = 1$. The overall fraction of cells released from the primary tumor is independent of $\sigma$ (Fig 7C) and mainly depends on the mean value of $\mu$ at a fixed value of $\Gamma$.

To measure the heterogeneity of the cells in the formed clusters, we calculated mean and standard deviation (std) of $\mu$ for each cluster size ($N$) at different values of $\sigma$. If all the cells distributed homogeneously into the clusters, we would expect that the mean and the std of $\mu$ for each cluster size ($N$) should be around 40 and $\sigma$ respectively. Instead, we observe a clear deviation of mean of $\mu$ from 40 depending on the cluster size ($N$) (see Fig 7D). This becomes more evident for larger values $\sigma$ (compare the results between $\sigma = 5$ and 10). The mean of $\mu$ for larger clusters ($N > 3$) is greater than that for smaller clusters and it drastically decreases for $N = 1$. We plot the distribution of $\mu$ for $N = 1$ in S11 Fig. The distribution clearly shifts towards lower value of $\mu$ for $N = 1$ with respect to the distribution of all the H cells. Our results show that the cells in smaller clusters ($N \leq 3$) are more epithelial-like compared to the cells in the larger clusters. We also observe a trend in std of $\mu$ with respect to $N$ (see Fig 7E). For smaller clusters ($N < 5$), the std of $\mu$ lies near the respective $\sigma$, as expected, but for larger clusters it is $< \sigma$.

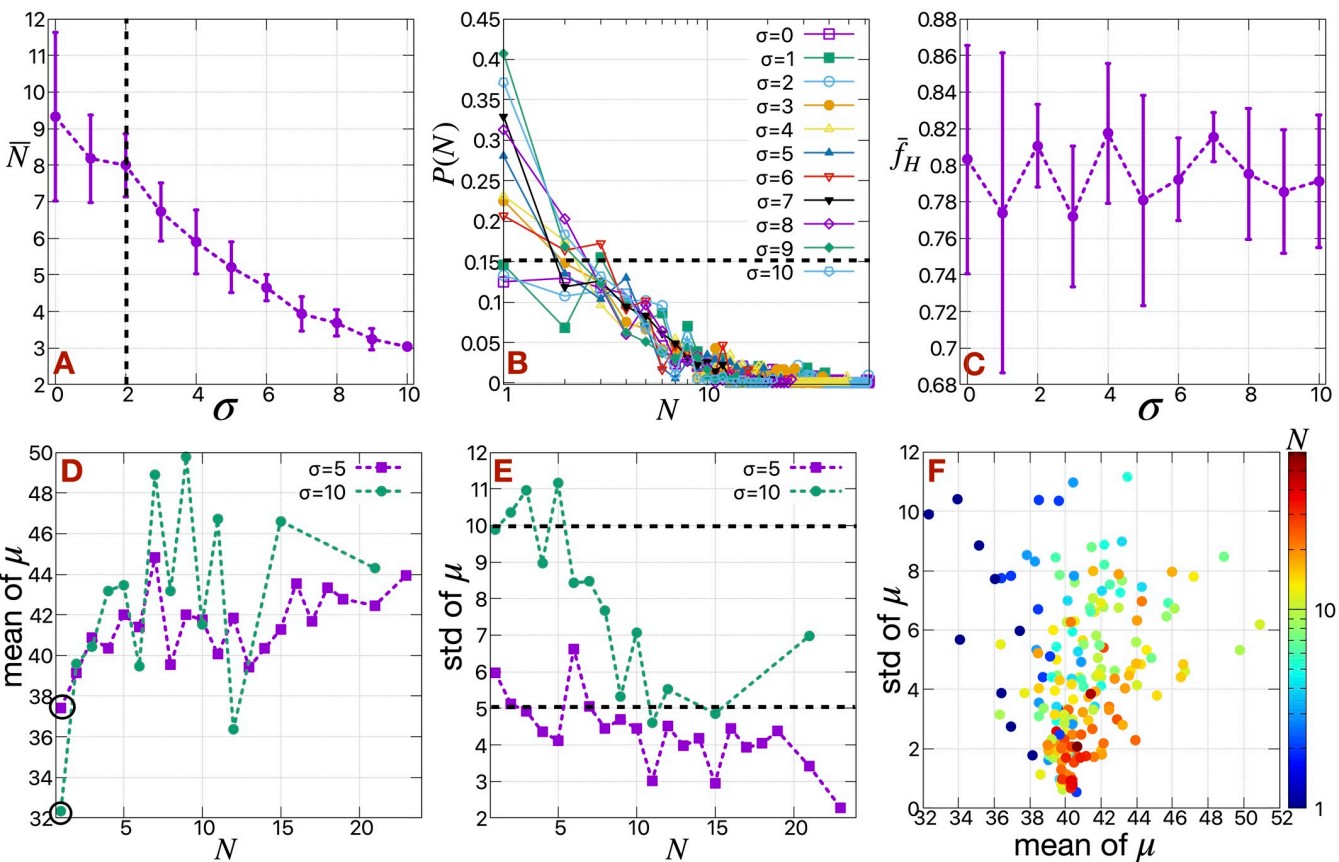

**Fig 7. Details of the clusters composed of a spectrum of EMT states.** (A) The mean cluster size ($\overline{N}$) of H cells as a function of $\sigma$, when motile forces of all the H cells are drawn from a normal distribution of mean 40 and standard deviation $\sigma$ at $\Gamma = 3$. (B) The corresponding cluster size distributions for different values of $\sigma$. The distributions are normalized such that $\sum P(N) = 1$. The N axis is in log scale. (C) The fraction of H cells ($\overline{f}_H$) released in the medium as a function of $\sigma$. We use standard deviations of the data as error bars. The (D) mean and (E) standard deviation (std) of $\mu$ for each cluster size N in case of $\sigma = 5$ and 10. (B) The heat map of cluster sizes (N) as function of mean and standard deviation (std) of motile forces ($\mu$). The N is in log scale.

We create a scatter plot in Fig 7F of the mean and std of $\mu$ for all N by collecting simulation data for all $\sigma = [0, 10]$. Cells with higher $\mu$ cluster together, and pull out small portion of other cells with lower $\mu$ from the tumor. These cells are often left behind when the differences in $\mu$ between the two group of cells are very large (as we already saw in case of binary $H_1$, $H_2$ cases previously). Due to this cluster breakup in the medium, smaller clusters are mainly composed of cells with lower motility, hence are relatively more epithelial-like and the larger clusters are mainly composed of cells with higher motility, and are relatively more mesenchymal-like. This can be a reason for CTCs to be relatively more mesenchymal compared to the primary tumor, as found in the case of Stage II-III breast cancer (GSE111842) [49, 50]. In general, it would be of interest to compare EMT-scores from clusters of different sizes taken from patient blood.

## Cell shape driven cluster size distribution

Apart from loss in adhesion between cells and gain in motility, the cell rigidity and cell shape also change during EMT. A cell shape driven rigidity transition has been observed in case of carcinoma [51]. To study this effect, we incorporate an additional term in the Hamiltonian of the system constraining the perimeters of the H cells, similar to the term related to constraining the area of the cells. We do not change the properties of the E cells.

We first note that in the absence of this perimeter constraint, the H cells attain a perimeter of mean value close to 25 (S12(A) Fig); there is also a long tail in the distribution of perimeters of H cells is due to a few cells at the boundary of the primary tumor, which cannot release (see S12(D) and S12(E) Fig). Interestingly, the mean cluster size ($\overline{N}$) decreases gradually with either an increase or decrease in the targeted perimeter ($p_0$), away from this value (see Fig 8A). In other words, when the cells on an average attain their targeted perimeters ($p_0$) such that $p_0 - \overline{p} \approx 0$ (or, the cortical tension $\lambda_p(\overline{p} - p_0)$ of the cells $\approx 0$), the large clusters are maintained, as shown in Fig 8B. When the $p_0$ is very large, the cells can not reach their targeted perimeters ($\overline{p} < p_0$), the cortical tension is negative, it is easier to break the clusters. The time series of $\overline{N}$ confirms this fact (see Fig 8C and 8D), initially released large clusters quickly break as they traverse the medium. The shape index/circularity of the cells increase/decrease with the increase in $p_0$ as shown in S13 Fig. At sufficiently high value of $p_0$, cells become extended in shape and can be thought of being relatively more mesenchymal; this gives rise to smaller values of $\overline{N}$.

In the other limit when $p_0$ is small ($< 24$), the cortical tension on the cells is positive ($\overline{p} > p_0$). The initially released clusters are larger compared to the larger $p_0$ limit, at a particular value of $\mu$ (see Fig 8C and 8D). But, they too eventually break and $\overline{N}$ again saturates to a smaller value compared to $p_0 = 25$. At very small value of $p_0$ ($< 20$), the cells become too rigid to release from the primary tumor.

At very high value of $p_0$ ($p_0 > 30$), the individual cells can actually start to break apart (see S13(G) and S13(K) Fig for $p_0 = 36$), which is an artifact of the Potts model [52]. We note that this does not appear to directly affect our results in the range of parameters that we have considered, as we do not see any significant change in mean cluster size ($\overline{N}$) as we increase $p_0$ for $p_0 > 30$ compared to at $p_0 = 30$. In any case, the non monotonic behavior of $\overline{N}$ with respect to $p_0$, is prominent in the range of $p_0 = [22, 30]$ which does not suffer from this potential complication.

## Conclusion and summary

In this article, we study the statistics of cell clusters arising from the dissemination of motile cells from a primary tumor composed of both motile and nonmotile cells. This is done within the framework of a computational approach based on the cellular Potts model formulation. The degree of mobility or equivalently the EMT status of the cells is modeled by two factors, namely the active motile forces of the cells ($\mu$) and cell-medium surface tension ($\Gamma$). The cells exit the primary tumor either as individuals or as clusters. The clusters can also break apart in the medium after disseminating from the primary tumor. Our main results concern identifying primary determinants of the cluster size distribution.

We found two distinct parameter regions according to the assumed $\Gamma$ value of the cells– (1) if $\Gamma \leq 0$, cells come out as individuals irrespective of $\mu$ and (2) for $\Gamma > 0$, the cells emerge as clusters with variable sizes depending on $\mu$. When a really large cluster emerges out from the tumor, it eventually breaks into smaller clusters. For each $\Gamma$ ($> 0$), there is a minimum value of $\mu$ ($\mu_{min}$) to release clusters, and the $\mu_{min}$ increases with the increase in $\Gamma$. The final distribution of cluster sizes depends on the ($\mu$, $\Gamma$). The probability of getting larger clusters is always small compared to that of smaller clusters, but experiments have shown that their higher metastatic potential can compensate for their smaller numbers. We find a relationship between mean cluster size ($\overline{N}$) and ($\mu > \mu_{min}$, $\Gamma > 1$), $\overline{N} \sim \Gamma^m/\mu^n$ with $m \approx 2.0$ and $n \approx 2.8$.

We also studied the effect of heterogeneity of cell types in the primary tumor on the resultant heterogeneity of cell types in clusters of different sizes. Cell types with different degrees of EMT can arise in the primary tumor depending on the nature of the EMT program. Without considering any explicit EMT dynamics, we assigned different $\mu$ values to different cells,

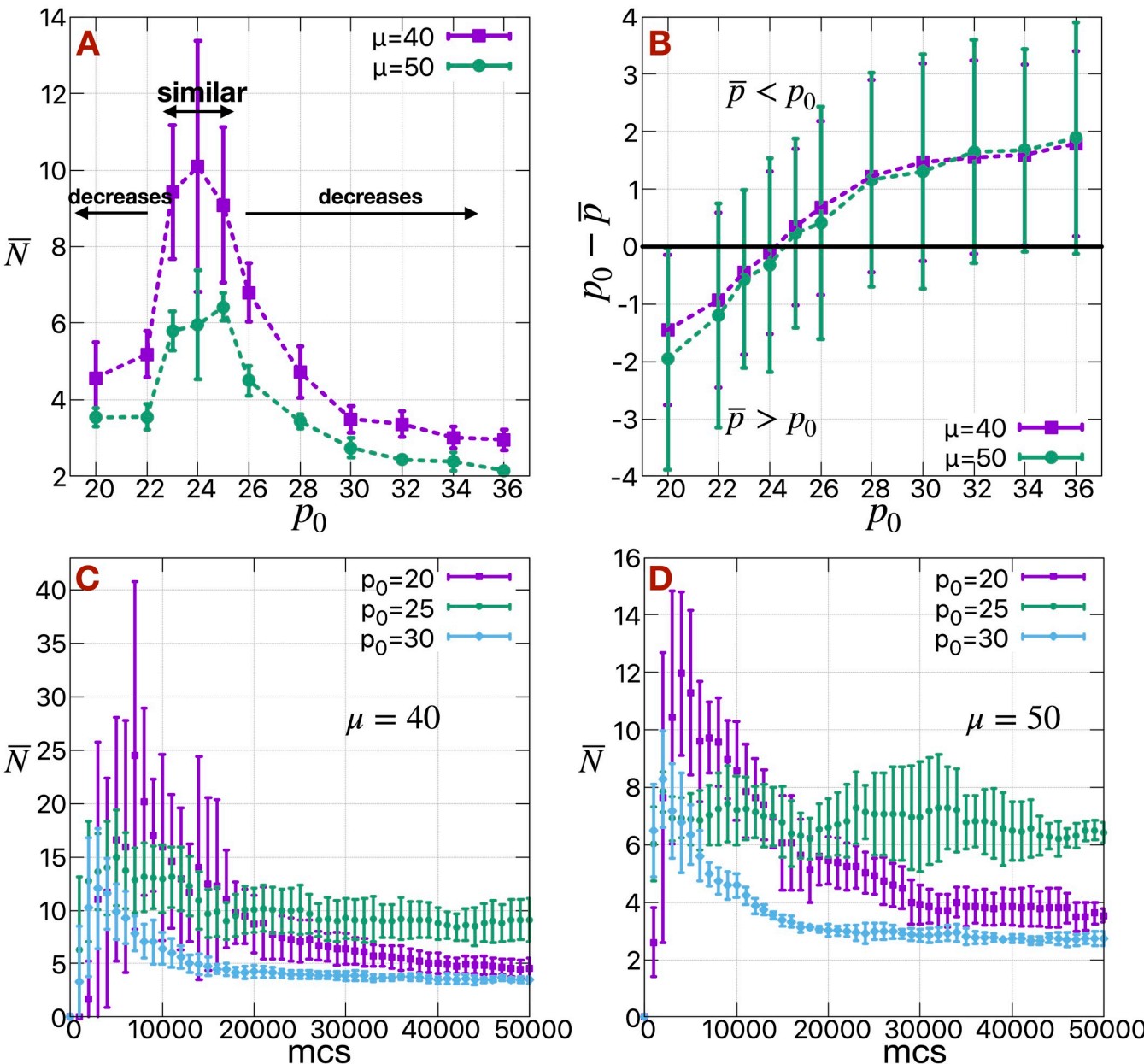

**Fig 8. Effect of cell shapes on cluster sizes.** (A) The mean cluster size ($\overline{N}$) of H cells as a function of targeted perimeter of the H cells ($p_0$) for different values of $\mu$ at $\Gamma = 3$ calculated at mcs = 50000. $\overline{N}$ is similar for $p_0 = [23, 25]$ and decreases gradually with either an increase or decrease in $p_0$. (B) The difference in targeted perimeter and mean obtained perimeter of H cells ($p_0 - \overline{p}$) as a function of $p_0$. The corresponding time series of mean cluster size ($\overline{N}$) of H cells for different values of $p_0$ at (C) $\mu = 40$ and (D) $\mu = 50$. We use standard deviations of the data as error bars.

mimicking the different EMT states of the cells in the primary tumor. We observe that a large variability in $\mu$ actually leads to smaller mean cluster size. Also, in a distribution of different cluster sizes arises from a tumor with fixed bulk EMT score (the mean of the distribution of $\mu$ for all the cells are fixed), the variability of $\mu$ is reduced in larger clusters compared to that in relatively smaller clusters. Moreover, the mean of $\mu$ in larger clusters is higher compared to that in relatively smaller clusters. As a consequence, the larger clusters are relatively more mesenchymal compared to smaller clusters.

Our methodology determines the cluster size distribution given a number of biophysical parameters governing the interaction of various EMT phenotypes. These results allow us to make a number of suggestions as to future experimental tests of our model. We imagine constructing tumor spheroids consisting of a mixture of epithelial cells and cells that have undergone some degree of EMT. This mixture can be constructed by using different subpopulations of given cell line, created by sorting cells for markers reflecting their position along the epithelial-mesenchymal axis. Alternatively, one can mix different cell lines. By using different mixtures, one can then vary the properties of the disseminating cells, for example their E-Cadherin levels which directly affect our adhesion parameter and/or the Vimentin levels which directly affects our motility difference parameter and the correlation time $\tau$. The correlation time could also be varied by changing the microstructure of the extracellular material through which the cells migrate [53]. Data from these studies could then be compared directly to our simulation predictions.

Of course, a more complete model (certainly one that might eventually be applied to *in vivo* situations) should include at least one additional biophysical process, namely cell division. Aside from the increased complexity of the model, doing this would require an understanding of the relative growth rates of different cell phenotypes as compared to the rates at which cells leave the tumor. It is true that cells that undergo EMT often significantly reduce their division rate (this is sometimes called "go versus grow") and if so, clusters should not have their size distribution dramatically altered by adding in division. It is also true, as seen by the initially larger clusters formed when starting from a larger tumor which break apart and reach the same dynamic steady-state distribution, that including cell division in the disseminated clusters might not in the end have a significant effect on the distributions. A full (and future) study of this issue would be needed to fully verify the conditions under which neglecting division events is a safe assumption.

One future extension of the current work would allow cells to undergo spontaneous phenotypic transitions as a function of their local environment. This would require adding EMT circuits to individual cells and coupling these to external chemical signals such as TGF-$\beta$ and cell-cell interactions such as the ones mediated by the Notch pathway [54]. A key issue here is the need for some degree of "phenotypic locking", that is the fact that transient signals seem capable of inducing semi-permanent changes in EMT status. For example, it has been argued in a mouse model of melanoma that cells undergo an EMT-like transition by being exposed to Notch ligands in the upper part of the skin, but that thereafter the cells retain their mesenchymal nature even as they migrate away from that specific region [55]. One possible source of this locking-in concerns epigenetic factors that alter the effective parameters of the responsible genetic circuit for EMT [56]. We hope to report soon on the effects of coupling such circuits to the motility mechanisms investigated here.

Although the study here has addressed only one aspect of the metastatic cascade, namely the dissemination of cells from the primary tumor, nevertheless we believe it addresses a critical issue. Experiments have clearly shown that clusters are major contributors to successful seeding of new lesions and hence their presence are predictors of poor prognosis. This effect is possibly due in part to enhanced survival of clusters in the bloodstream but the major reason for this is that clusters are more likely carriers of hybrid E/M cells with enhanced propensities to initiate new growth in foreign micro-environments [20, 23, 24]. Understanding how phenotypic diversity eventually leads to the creation of these dangerous actors will undoubtedly prove useful in our ongoing battle with this most deadly aspect of cancer progression.

## Supporting information

**S1 Video. The dynamics of the cells and dissemination from a primary tumor in case of $\mu$ ($H_1$) = 40 and $\mu(H_2)$ = 35.**
(MOV)

**S2 Video. The dynamics of the cells and dissemination from a primary tumor in case of $\mu$ ($H_1$) = 40 and $\mu(H_2)$ = 20.**
(MOV)

**S1 Table. The details of the parameters for different figures in main text and supporting information.** We use surface tensions ($\gamma$) instead of contact energies ($J$), defined as $\gamma_{Em} = J_{Em} - J_{EE}/2$, $\gamma_{Hm} = \Gamma = J_{Hm} - J_{HH}/2$. The uncommon ones which are different from the standard parameters used in most of the simulations are colored in red. We mainly focus on the two parameters $J_{Hm}$ (or in other words $\gamma_{Hm} = \Gamma$) and $\mu$ related to H cells in this article.
(TIFF)

**S1 Fig. The effect of $J_{EH}$ and $T_H$ on clusters in case of diffusive motility.** The (A) mean cluster size ($\overline{N}$) of H cells and (B) fraction of H cells ($\overline{f}_H$) released in the medium as a function of $J_{EH}$ at $T_H$ = 6. The (C) mean cluster size ($\overline{N}$) of H cells and (D) fraction of H cells ($\overline{f}_H$) released in the medium as a function of $T_H$ at $J_{EH}$ = 30. We use standard deviations of the data as error bars.
(TIFF)

**S2 Fig. Representative velocity map of the cells at different time (mcs) showing the releasing and breaking of H cell clusters.**
(TIFF)

**S3 Fig. The time series of mean cluster size.** The time series of mean cluster size ($\overline{N}$) of H cells for different values of $\mu$ at (A) $\Gamma$ = 4, (B) $\Gamma$ = 3, (C) $\Gamma$ = 2 and (D) $\Gamma$ = 1.
(TIFF)

**S4 Fig. The effect of $T_H$ and $\lambda_a$ on clusters.** The (A) mean cluster size ($\overline{N}$) of H cells and (B) fraction of H cells ($\overline{f}_H$) released in the medium as a function of $T_H$ at $\lambda_a$ = 1. The (C) mean cluster size ($\overline{N}$) of H cells and (D) fraction of H cells ($\overline{f}_H$) released in the medium as a function of bulk modulus of H cells $\lambda_a$ at $T_H$ = 1. In all cases $\mu(H)$ = 40 and $\gamma_{Hm} = \Gamma$ = 3. We use standard deviations of the data as error bars.
(TIFF)

**S5 Fig. The effect of $\gamma_{Em}$ and $T_E$ on clusters.** The (A) mean cluster size ($\overline{N}$) of H cells and (B) fraction of H cells ($\overline{f}_H$) released in the medium as a function of $\gamma_{Em}$ at $T_E$ = 1. The (C) mean cluster size ($\overline{N}$) of H cells and (D) fraction of H cells ($\overline{f}_H$) released in the medium as a function of $T_E$ at $\gamma_{Em}$ = 19. In all cases $\gamma_{Hm} = \Gamma$ = 3, $T_H$ = 1 and $\mu(H)$ = 30 and 40. Representative snapshots of the primary tumor at final step (mcs = 50000) of simulation after most of the H cells are released for different values of $\gamma_{Em}$ = (E) 19, (F) 14, (G) 9 and (H) 4 at $T_E$ = 1. We use standard deviations of the data as error bars.
(TIFF)

**S6 Fig. The effect of $\tau$ on clusters.** The (A) mean cluster size ($\overline{N}$) of H cells and (B) fraction of H cells ($\overline{f}_H$) released in the medium as a function of $\tau$ for different values of $\mu$ at $\Gamma$ = 3.
(TIFF)

 

**S7 Fig. Cluster size distributions as a function of motile forces and cell-medium surface tension.** Cluster size distributions ($P(N)$) of H cells for different values of $\Gamma$ = (A) 4, (B) 3, (C) 2, (D) 1, (E) 0 and (F) −1. The distributions are normalized such that $\sum P(N) = 1$. The $N$ axes are in log scale.
(TIFF)

**S8 Fig. The effect of larger system size.** The mean cluster size ($\overline{N}$) of H cells at (A) $\mu$ = 40, $\Gamma$ = 3 and (A) $\mu$ = 40, $\Gamma$ = 2 for different size of the tumor with total number of cells 517 and 2109 composed of 60% nonmotile E cells and 40% motile H cells. We use standard deviations of the data as error bars. (C) and (D) The corresponding cluster size distributions ($P(N)$) of H cells. The distributions are normalized such that $\sum P(N) = 1$. The $N$ axes are in log scale.
(TIFF)

**S9 Fig. The mean cluster size in case of $H_1$ and $H_2$ cells different in their $\mu$.** (A) The mean cluster size ($\overline{N}$) of H cells as a function of $\mu(H_2)$ in case of two kinds of H cells different in their motile forces at a fixed $\Gamma$ = 3 for different values of $\mu(H_1)$. (B) The corresponding difference of $\overline{N}$ ($\Delta\overline{N} = \overline{N}(H_2) - \overline{N}(H_1)$) as a function of $\Delta\mu = \mu(H_2) - \mu(H_1)$. (C) The mean cluster size ($\overline{N}$) of H cells as a function of $\gamma(H_2)$ in case of two kinds of H cells different in their cell-medium surface tensions ($\gamma$) for $\mu(H_1) = \mu(H_2)$ = 30 and 40 at $\Gamma(H_1)$ = 3. (D) The corresponding difference of $\overline{N}$ ($\Delta\overline{N} = \overline{N}(H_2) - \overline{N}(H_1)$) as a function of $\Delta\Gamma = \Gamma(H_2) - \Gamma(H_1)$. We use standard deviations of the data as error bars.
(TIFF)

**S10 Fig. The cluster size distributions in case of $H_1$ and $H_2$ cells different in their $\Gamma$.** The cluster size distributions ($P(N)$) of H cells in case of two kinds of H cells different in their cell-medium surface tensions ($\Gamma$) at a fixed (A-D) $\mu$ = 40 and (E-H) $\mu$ = 30 for different values of $\Delta\Gamma = \Gamma(H_2) - \Gamma(H_1)$ by changing $\Gamma(H_2)$ keeping fixed $\Gamma(H_1)$ = 3. At each $N$, $P(N)$ is splitted into three bars. Red bars represent the clusters consist of only $H_1$ cells, blue bars represent the clusters consist of only $H_2$ cells and green bars represent the heterogeneous clusters consist of $H_1$ and $H_2$ cells. The distributions are normalized such that $\sum P(N) = 1$.
(TIFF)

**S11 Fig. The distribution of motile forces of H cells in presence of spectrum of spectrum of EMT states.** The distribution of motile forces ($\mu$) of all the H cells for (A) $\sigma$ = 5 and (B) $\sigma$ = 10, when motile forces of all the H cells are drawn from a normal distribution of mean 40 and standard deviation $\sigma$. (C-D) The similar distributions of $\mu$ for single H cells ($N$ = 1) in the medium.
(TIFF)

**S12 Fig. The distribution of perimeters.** The distribution of perimeters of all the H cells in case of (A) the Hamiltonian without perimeter constraint, the Hamiltonian with perimeter constraint and targeted perimeter (B) $p_0$ = 24, (C) $p_0$ = 25. (D) Representative snapshot of the primary tumor at final step (mcs = 50000) of simulation after most of the H cells are released. (E) The heat map of perimeters of all the cells in the primary tumor at the final step.
(TIFF)

**S13 Fig. The effect of perimeter on the clusters.** The mean (A) area ($\overline{a}$), (B) shape index $\left(\overline{\frac{p}{\sqrt{(a)}}}\right)$ and (C) circularity $\overline{\left(4\pi\frac{a}{p^2}\right)}$ of all the H cells as a function of targeted perimeter ($p_0$). We use standard deviations of the data as error bars. The representative snapshots of the single H

cells and small H cell clusters at different values of $p_0$ in case of (D-G) $\mu = 40$ and (H-K) $\mu = 50$. (TIFF)

## Acknowledgments

We acknowledge useful discussions with Mohit Kumar Jolly and Dapeng (Max) Bi.

## Author Contributions

**Conceptualization:** Mrinmoy Mukherjee, Herbert Levine.

**Formal analysis:** Mrinmoy Mukherjee.

**Funding acquisition:** Herbert Levine.

**Investigation:** Mrinmoy Mukherjee.

**Methodology:** Mrinmoy Mukherjee.

**Supervision:** Herbert Levine.

**Visualization:** Mrinmoy Mukherjee.

**Writing – original draft:** Mrinmoy Mukherjee.

**Writing – review & editing:** Mrinmoy Mukherjee, Herbert Levine.

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
