## [Decision Letter · Decision Letter 0]

26 Jul 2021

Dear Dr. Mukherjee,

Thank you very much for submitting your manuscript "Cluster Size Distribution of Cells Disseminating from a Primary Tumor" for consideration at PLOS Computational Biology.

As with all papers reviewed by the journal, your manuscript was reviewed by members of the editorial board and by several independent reviewers. In light of the reviews (below this email), we would like to invite the resubmission of a significantly-revised version that takes into account the reviewers' comments.

In particular, as the reviewers comment, how well do the results agree with biological observations, and can they be tested experimentally? Are there experimental observations that the model can help explain, or are there key discrepancies with data that warrant further research? Apart from the detailed comments by reviewers #2 and #3, the biological insight coming from your model and the biological interpretation of your results will be key for deciding on the manuscript's suitability for PLOS Computational Biology. The reviewers further remark that the figures are difficult to interpret. Ensure that figures can be read  fairly independently from the main text - see for example the labels "similar", "increasing" in Fig. 8 and the detailed remarks of reviewer #1.

We cannot make any decision about publication until we have seen the revised manuscript and your response to the reviewers' comments. Your revised manuscript is also likely to be sent to reviewers for further evaluation.

Sincerely,

Roeland M.H. Merks, Ph.D

Associate Editor

PLOS Computational Biology

Douglas Lauffenburger

Deputy Editor

PLOS Computational Biology

Reviewer's Responses to Questions

**Comments to the Authors:**

Reviewer #1: In this study the authors carried out numerical simulations of cellular Potts model to investigate cluster size of disseminating cells from primary tumor. It was confirmed that the cluster size increases for larger adhesion while decreases for larger motility, for which scaling relation were determined. Mixture of different cell types were also investigated, where combination of different cell types can facilitate larger cluster size.

Overall, the author has shown many results in various simulation settings, but the analysis is rather descriptive for each setting and the results are not very surprising. I'm not sure if the results are general and / or insightful enough to understand the actual process of metastasis.

1. Some figures are too small and messy, due to which I’m afraid that I misunderstand the results, e.g., Figs. 3C,D, 4F, 5, 6.

2. Comparison with experimental data are scattered in the main text and are not sufficiently addressed. For example, I’m not sure whether experimental data shown in Fig. 3f are similar to numerical results. The authors should make a more detailed discussion of the interpretation of the obtained results in the light of experimental observation. This will help the readers understand the motivation and purpose of this study.

3. What is the biological meaning of the parameter tau? Is it possible to experimentally disturb it?

4. The author numerically determined powers of scaling behaviors of N and f on mu and Gamma. Are these values are universal, in other words, simulation with other model schemes can also reproduce the same results? Is it possible to theoretically derive these powers?

5. How mu_min scales with Gamma? Using mu – mu_min instead of mu seems more natural choice (I might be wrong).

6. Fig. 7A and L313. The authors claim “N remains almost constant up to sigma = 2, ..”, but it is unclear this statement is valid from Figs 7A and B.

7. Fig. 7D and E: adding error bars is preferable.

8. The section “Cell shape driven cluster size distribution” seems interesting. Are the results related with jamming transition reported in Bi, et al. Nat. Phys. (2015)?

9. Typos: L226 “fall rapidly fall as N ..”, L420 “Nevertheless” N is capital.

Reviewer #2: Review is uploaded as an attachment.

Reviewer #3: The manuscript presents a detailed study of a cellular Potts model with the aim of chracterizing clusters of metastatic cells in terms of the simulted EMT state of the individual cells comprising the cluster. Understanding how the phsyical organization of metastatic cells depends on the gene regulated phenotype of the individual cells undergoing EMT is an interesting topic worth investigating. The present computational study could guide further experimental studies. In this respect, I woul encourage the authors to discuss in the conclusions possible avenues for experimental investigation of the same problem. In general the manuscript is well written and I would thus suggest publication.

I have some minor issues that should be fixed before acceptance:

The analysis presented in Fig. 3 should be improved. The author state that the fit in Fig. 3A yields an exponent of 1.98 +- 0.01. Yet ,the fit is performed over three points with huge error bars. How is it possible that the error bar on the exponent is so tiny. Was the error on the data included in the estimate of the error bar on the fit (my guess is that the authors reported what the fitting routine gave). Also, it is not clear, here and in other instances what the error bar actually is. It is the SD or the SE on the mean? Please provide the definition of the error bars in all the captions. I have similar considerations for the fit p=0.999 +- 0.001 in Fig. 3c. It is clear that a line fits the data, but I do not believe the tiny error bar. Since the author are fitting multiple curves in a parameter dependent way, the most correct way to proceed is to fit collectively all the curves with a single scaling function with unknown parameters. In this way, all the parameters are fit simultaneously over multiple curves.

**Have the authors made all data and (if applicable) computational code underlying the findings in their manuscript fully available?**

Reviewer #1: **No: **I did not notice code and detailed data list. The results themselves are reproducible by the given explanation.

Reviewer #2: **No: **

Reviewer #3: Yes

PLOS authors have the option to publish the peer review history of their article (what does this mean?). If published, this will include your full peer review and any attached files.

Reviewer #1: No

Reviewer #2: No

Reviewer #3: No
---

## [Decision Letter · Decision Letter 1]

27 Sep 2021

Dear Dr. Mukherjee,

Thank you very much for submitting your manuscript "Cluster Size Distribution of Cells Disseminating from a Primary Tumor" for consideration at PLOS Computational Biology. As with all papers reviewed by the journal, your manuscript was reviewed by members of the editorial board and by several independent reviewers. The reviewers appreciated the attention to an important topic. Based on the reviews, we are likely to accept this manuscript for publication, providing that you modify the manuscript according to the review recommendations.

Sincerely,

Roeland M.H. Merks, Ph.D

Associate Editor

PLOS Computational Biology

Douglas Lauffenburger

Deputy Editor

PLOS Computational Biology

[LINK]

Reviewer's Responses to Questions

**Comments to the Authors:**

Reviewer #1: The authors answered my comments and questions, and the revised manuscript is much improved. I now agree the manuscript meets publication from the PLOS Compt. Biol.

Reviewer #2: The authors satisfactorily answered my comments, except the one about fragmentation, which has been overlooked. The authors claimed that they work in a region of parameter space where fragmentation events are very rare, as they observe fragmentation at p0>30. However, this represents 37.5% of the range they scanned ([20,36]).

As this has been shown recently (M. Durand, PLOS Comp. Biol. 17, e1008576, 2021), apparition of fragments can significantly affect the results of CPM simulations. Therefore, the results presented in Fig. 8, S12 and S13 should be presented with proper warnings, at least.

Reviewer #3: The authors modified the paper in order to respond to the remarks of the referees. I am satisfied by the response.

**Have the authors made all data and (if applicable) computational code underlying the findings in their manuscript fully available?**

Reviewer #1: Yes

Reviewer #2: Yes

Reviewer #3: Yes

PLOS authors have the option to publish the peer review history of their article (what does this mean?). If published, this will include your full peer review and any attached files.

Reviewer #1: No

Reviewer #2: No

Reviewer #3: No

Figure Files:

Data Requirements:

Reproducibility:

References:

---

## [Decision Letter · Decision Letter 2]

25 Oct 2021

Dear Dr. Mukherjee,

We are pleased to inform you that your manuscript 'Cluster Size Distribution of Cells Disseminating from a Primary Tumor' has been provisionally accepted for publication in PLOS Computational Biology.

Best regards,

Roeland M.H. Merks, Ph.D

Associate Editor

PLOS Computational Biology

Douglas Lauffenburger

Deputy Editor

PLOS Computational Biology

Reviewer's Responses to Questions

**Comments to the Authors:**

Reviewer #2: The authors satisfactorily answered all my comments, and the revised manuscript is now suitable for publication in PLOS Comput. Biol.

**Have the authors made all data and (if applicable) computational code underlying the findings in their manuscript fully available?**

Reviewer #2: Yes

PLOS authors have the option to publish the peer review history of their article (what does this mean?). If published, this will include your full peer review and any attached files.

Reviewer #2: No

---

## [Editor Report · Acceptance letter]

4 Nov 2021

PCOMPBIOL-D-21-00759R2 

Cluster Size Distribution of Cells Disseminating from a Primary Tumor

Dear Dr Mukherjee,

I am pleased to inform you that your manuscript has been formally accepted for publication in PLOS Computational Biology. Your manuscript is now with our production department and you will be notified of the publication date in due course.

With kind regards,

Livia Horvath
